# Private Keeping of Dangerous Wild Animals in Great Britain

**DOI:** 10.3390/ani14101393

**Published:** 2024-05-07

**Authors:** Chris Draper, Chris Lewis, Stephanie Jayson, Frankie Osuch

**Affiliations:** 1Performing Animal Welfare Society, Galt, CA P.O. Box 849, USA; 2Born Free Foundation, Frazer House, 14 Carfax, West Sussex RH12 1ER, UK; clewis@bornfree.org.uk (C.L.);; 3Farthings Veterinary Group, Farthings Hill, Guildford Road, West Sussex RH12 1TS, UK

**Keywords:** dangerous wild animals, animal welfare, private keeping, exotic pet, public safety

## Abstract

**Simple Summary:**

In Great Britain, private animal keepers such as pet owners and farmers require a licence from their local government authority to keep wild animals of species formally categorised as “dangerous”. No central record of these licences is maintained by the national Governments in Britain, and the last formal examination of the situation was published more than 20 years ago. We examined the scale and scope of private keeping of dangerous wild animals in England, Scotland, and Wales in 2020. We found a total of 3950 individual dangerous wild animals kept by 210 licensees across more than one-third of British local authorities. While overall numbers of licensed dangerous wild animals have declined over the preceding 20 years, there have been notable increases in the keeping of some taxa such as wild cats, venomous snakes, and crocodilians. There is evidence that the average relative cost to obtain a licence to keep a dangerous wild animal has decreased since the earlier study, and that local authorities with licensed animals have lower licensing fees than those without. We discuss the current system of licensing with a view to making recommendations for improvement.

**Abstract:**

We analysed the licences issued by British local government authorities under the Dangerous Wild Animals Act 1976, which regulates the private keeping of wild animals categorised as “dangerous”, to assess the scope and scale of private keeping of dangerous wild animals in Great Britain. Results are compared with historical data from England and Wales, showing that there has been an overall decrease both in the total population of dangerous wild animals privately kept under licence and the number of licences, resulting primarily from a decrease in the farming of wild boar and ostrich, and from certain other species no longer requiring a licence to be kept. Nonetheless, the private keeping of dangerous wild animals remains prevalent, with a total population of 3950 animals kept under licence, and at least one-third of local authorities in Britain licensing keepers of one or more such animals. The population of non-farmed dangerous taxa has increased by 59% in 20 years, with notable increases in crocodilians (198%), venomous snakes (94%), and wild cats (57%). We present evidence that the average cost of a licence to keep dangerous wild animals has fallen over time, and that there is a negative association between cost and licensing. The current schedule of species categorised as dangerous is compared to a formally recognised list of species kept in zoos assessed by risk to the public. Problems with the legislation, enforcement of the licensing system, and animal welfare for privately kept dangerous wild animals are identified and discussed.

## 1. Introduction

The keeping by private individuals (such as pet owners and farmers) of wild animals categorised as dangerous has required licensing in Great Britain since the enactment of the Dangerous Wild Animals Act 1976 (henceforth “the Act” or “DWAA”) [1]. The Act does not apply to dangerous wild animals kept in zoos, circuses, laboratories, or by licensed pet sellers. The original objective of the Act was to “regulate the keeping of certain … dangerous wild animals” to avoid these animals being kept in cruel and unsuitable conditions and to avoid dangerous animals being kept that could be a risk to other people [2]. A key aim was to protect the safety of the general public, rather than the keepers of dangerous wild animals and their households [3]. The Act was introduced following a number of widely publicised incidents involving escapes and attacks on members of the public by large wild cats; and a growing trend in large cats being advertised for sale and bought as pets, fuelled by a surplus of these animals from zoos and safari parks [2,3]. The general intent of the Act was that, in future, the keeping of dangerous wild animals by private individuals should be made a wholly exceptional circumstance [2].

Wild animal taxa considered to be dangerous and for which a licence is required are listed in Schedule S, Section 7 of the DWAA. Unusually, the criteria for inclusion in the Schedule are not clear and the Act does not offer a definition or threshold for “dangerous”. In contrast to animals in zoos in Great Britain, for example, which are categorised into three risk levels depending on the danger they present to the public, animals listed in the Schedule of the DWAA are not assigned a risk rating; they are simply “dangerous” [4]. While common names are provided in the Schedule to the Act, it is specifically stated that common names are provided by way of explanation only, and that only scientific names should be taken into account in the case of dispute or legal proceedings.

Licences are issued by local authorities in England, Scotland and Wales. DWAA licences are issued for a period of two years; previously, licences were renewed annually [5]. The Act requires applicants to specify the species of animal and the number of each species to be kept under licence, the location of the premises where the animals are to be kept, and that the applicant is over the age of 18 and has not previously been disqualified from keeping dangerous wild animals under the Act. Additionally, local licensing authorities must be satisfied that the granting of a licence is not contrary to public welfare in terms of safety, nuisance, or otherwise; the applicant is a suitable person who owns and possesses the animal; the enclosure of the animal prevents its escape, but is also suitable for the animal in terms of its construction, size, environmental conditions, sanitation, and capability to allow the animal adequate exercise; food, drink, and bedding provision are suitable and adequate; a vet, acting on behalf of the licensing authority, has provided a report stipulating the premises have been inspected and approved; there are appropriate precautions in place in case of fire and the emergence of infectious disease; and an active liability insurance policy has been acquired by the applicant in case of any damage caused by the animal [1]. Certain species which are listed in the Act’s Schedule are also covered by the requirements of the Convention on International Trade in Endangered Species of Wild Fauna and Flora (CITES) [6]. Although not explicitly stated within the Act, proof of legal possession would theoretically involve the presentation of suitable CITES permits where applicable.

The species listed in the Schedule have changed over time. In 1981, rheas (*Rhea* spp.) were removed from the Schedule and 12 taxa were added [7]. In 1984, racoon dogs (*Nyctereutes procyonoides*) were removed from the Schedule, while New World primates (except marmosets) and certain mustelids, snakes, spiders, scorpions, and hybrids were added [7]. In 2007, a wide range of taxa were removed from the Schedule, including certain smaller primates, sloths, some large rodents, procyonids, binturong (*Arctictis binturong*), cat hybrids which are predominantly domestic cats, hyraxes, certain camelids, emus (*Dromaius novaehollandiae*), sand snakes (*Psammophis* spp.), mangrove snakes (*Boiga* spp.), and the Brazilian wolf spider (*Phoneutria* spp.) [8].

Although the avoidance of cruelty was cited as among the reasons for introducing the Act, animal welfare aspects of the Act were considered secondary to public safety, and the animal welfare provisions within the Act are fairly rudimentary [2,3,9]. Vertebrate animals held under a DWAA licence are afforded welfare protection under other legislation, notably the Animal Welfare Act 2006 in England and Wales, and the Animal Health and Welfare (Scotland) Act 2006 in Scotland [10,11]. Those animals that are kept or trained for exhibition in England, for example for use in films or TV, may also fall within the scope of the Animal Welfare (Licensing of Activities Involving Animals) (England) Regulations 2018 (LAIA), for which welfare guidance exists [12]. However, the welfare requirements and guidance under these pieces of legislation are both broad and limited; generally, they lack specific detail relevant to the species scheduled in the DWAA [13]. Greenwood et al. [3] recommended that public guidance on the minimum standards of care required for commonly kept DWAA scheduled species was needed; however, this has not been fully addressed today, with only the Scottish Government providing guidance on the keeping of dangerous wild animals, which includes the “needs of the species considered and the requirements they [owners] may have to meet” [14]. A Code of Practice for the welfare of privately kept primates (a commonly kept taxon that includes numerous species on the Schedule of the DWAA) exists for England and Wales [15]. In 2014, the British Veterinary Association (BVA) produced a standard inspection template which asks a number of husbandry- and welfare-related questions which aimed to “ensure conformity in DWA licence inspections” [16]. In 2020, the British Veterinary Zoological Society (BVZS) highlighted its concerns about the welfare of dangerous wild animals in private ownership, citing the high proportion of reptiles presenting to veterinary practices with husbandry-related diseases and that providing good husbandry for venomous animals is made more difficult by the challenges in handling and managing these animals safely [17]. Other potential animal welfare problems in private keeping include those arising from isolation or restricted social groupings in social species [18,19,20]. The impact of the private keeping of dangerous wild animals on animal welfare has not been specifically examined, but it is likely to share problems with other private keeping situations.

The last substantive examination of the Act was undertaken in 2000, when a review of the Act was commissioned by the Department of Environment, Food and Rural Affairs (Defra) to determine its effectiveness [3]. A survey of local authorities was conducted, revealing that 375 DWAA licences were issued by 205 local authorities (52.8%) in England and Wales in 2000, for a total of 11,878 animals. The average annual licensing cost was GBP 131.05, with fees ranging from GBP 25 to GBP 525, plus veterinary inspection fees. At the time, farmed species (ratites, wild boar (*Sus scrofa*), bison (*Bison bison*), and guanaco (*Lama guanicoe*)) represented 88.8% of all licensed animals. It is highly likely that these figures underestimated the number of dangerous wild animals being kept by private individuals, as it is thought that a significant proportion of dangerous wild animal keepers in Great Britain do not apply for a DWAA licence [3,21]. The review of Greenwood et al. [3] concluded with a series of wide-ranging recommendations to resolve problems identified with the Act, including suggested changes to the Act, Schedule, and improved guidance.

More than twenty years after the review by Greenwood et al. [3], this paper aims to investigate the scale of dangerous wild animal keeping under licence in Great Britain in 2020. This includes an examination of changes in keeping over time, discussing species identification in the context of the Act, reviewing the effectiveness of the current legislation in protecting animal welfare and public safety, and analysing the influence of licensing fees.

## 2. Materials and Methods

On 18 March 2020, the number of DWAA licences and the details of animals kept under each licence were requested by email under the Freedom of Information Act 2000 from all 371 local licensing authorities in Great Britain (192 district councils, 109 unitary authorities, 36 metropolitan councils, 32 London boroughs, the City of London, and the Isles of Scilly Council).

Several local authorities which responded to our request for information and presented in Section 3.1 and Section 3.2 have merged with other local authorities, including their website and licence fees. However, a number still responded independently of each other to Freedom of Information requests or reported DWAA licences for each previously recognised local authority area. These are included separately for increased accuracy resulting in a difference of four between the total local authorities reported in Section 3.1 and Section 3.4.

Information from local authority responses was collated including licensing authority, number of licensed premises within each local authority, taxonomic classification of species kept (from class to species), and total number of animals by species under each licence.

Animals under licence were identified on the basis of reported binomial and/or common name. In cases where species could not be identified on the basis of the information in the response, local authorities were contacted to provide additional information. For consistency, reported binomials were checked for validity against the Integrated Taxonomic Information System (ITIS) (http://www.itis.gov, accessed on 1 June–15 August 2020) and amended accordingly. Any reported subspecies were checked for validity against ITIS, while analysis was performed at the species level (where possible). Where responses reported licences for species which are not on the DWAA Schedule, these were noted but removed prior to analysis.

The information regarding licensing fees was obtained from each local authority’s website. The following was recorded: (i) the cost to apply for a licence under the DWAA; (ii) whether the veterinary fee was included in the application fee; (iii) the veterinary inspection fee, where stated; and (iv) any additional charges per animal and/or species. Where the application fee was not stated, an email was sent to the local authority requesting this information. Non-respondents were called once using the telephone number provided on their website. If it was unclear whether the fee stated did or did not include the veterinary inspection fee, the local authority was contacted for clarification.

Where veterinary inspection fees were included in the stated licence application cost, and the vet fee was known, it was subtracted from the stated licence application cost. Data were omitted from the analysis if the vet fee was included in the stated cost, but the vet fee was unknown, or if the local authority stated that there was no set fee for a licence. Historical annual licensing costs were obtained from Greenwood et al. [3], adjusted using the 2021 Bank of England “Inflation Calculator”, and doubled to permit comparison with the cost of current biennial licences [22].

Data on group sizes were obtained from Jones et al. [23]. Data were analysed in R, version 4.3.1, with the use of the dplyr and tidyverse packages. Details of the tests are provided in the Results section. The significance level for all tests was *p* < 0.05.

## 3. Results

### 3.1. Licences Issued

Of 371 local licensing authorities, 364 responded to the request for information. Of those responding, 129 (35.4%) reported one or more current DWAA licence: 114 local authorities in England, 6 in Wales, and 9 in Scotland. For authorities reporting licences, the number of licensees ranged from 1 to 8 (mean = 1.6; median = 1).

A total of 210 licensed keepers were reported across Great Britain (Table 1).

The majority of licensed keepers (*n* = 196) kept only one taxonomic class of dangerous wild animal: 14 licensed keepers kept dangerous wild animals from two classes, while just 1 licensed keeper kept three classes of dangerous wild animal (arachnid, mammal, and reptile).

### 3.2. Licensed Animals

A total of 3950 individual animals were reported as being kept under a DWAA licence (Table 1). Numbers of individual dangerous wild animals kept by each licensed keeper ranged from 1 to 410 (mean = 18.8; median = 5) (Table 2). Most licensees kept multiple individual animals, as only 32 keepers (15.2%) were licensed for a single individual animal. Twenty percent of licensed keepers kept ≥ 20 individual animals.

Across Great Britain, the greatest number of licences were for servals (*Leptailurus serval, n =* 34), hybrid wild cats (“savannah” and “Bengal” cats, *n* = 27), rattlesnakes (*Crotalus* spp., *n* = 26), and ring-tailed lemurs (*Lemur catta*, *n* = 23) (Appendix A).

Valid subspecies were reported for 76 animals (including 60 snakes) across 19 licences.

#### 3.2.1. Birds

In total, 334 birds were reported across 19 licences. Both species of bird requiring a DWAA licence were kept in Great Britain: 98.8% (*n* = 330) were ostriches (*Struthio camelus*), plus four individual cassowaries (*Casuarius* spp.) kept by one licensed keeper.

[The family *Casuariidae* (“Cassowaries”) is listed on the Schedule and is understood to be distinct from *Dromaiidae* (including emus), despite some sources treating them as a single family [24]].

#### 3.2.2. Invertebrates

A total of 549 invertebrates were reported across eight licences, comprising 332 scorpions kept across seven licences and four licences with an approximate total of 217 spiders. The numbers of individual invertebrates kept ranged from two to four hundred and ten (mean = 68.6, median = 7). A total of 12 species could be identified from the information supplied.

#### 3.2.3. Mammals

A total of 2295 individual mammals were kept under 138 licences. Wild boar was the most prevalent species, accounting for 45% (*n* = 1034) of the mammal population, and consequently the most commonly kept mammals belonged to Artiodactyla—1564 individuals, followed by Carnivora—398, and Primates—273. A total of 80% of all carnivores kept under licence were felids, while 55% of all primates kept under licence were lemurs. A total of 75 species and three subspecies were reported.

Primates were kept under 41 licences across 37 local authorities. A total of 25 primate species were kept (excluding hybrids and species that could not be identified). The mean number of individuals kept by species was 3.389 (SD = 2.9), and there were 20 instances across 12 licences where only one individual of the species was held under that licence. Of those 12 licences, there were 8 licences where the animal was the only individual held under that licence. In total, 29% of licensed primate keepers kept a single representative of a species.

A total of 110 mammals were reported to be hybrids. Of these, 105 were felids—65 “savannah” cats (domesticated cat, *Felis catus* x serval, *Leptailurus serval*), 38 “Bengal” cats (domesticated cat, *F. catus* x Asian leopard cat, *Prionailurus bengalensis*), and two ligers (lion, *Panthera leo* x tiger, *Panthera tigris*). Almost one-third of individual felids kept under licence were hybrids.

#### 3.2.4. Reptiles

Dangerous reptiles were reported as being kept by 61 licensees.

In total, 43 licences holding 508 venomous snakes were reported. Of these, 60 animals were identified to the subspecies level, totalling 22 taxa.

A total of 27 licensees held 158 crocodilians. Venomous lizards were held under 16 licences and individual animals totalled 106.

#### 3.2.5. Changes over Time

Previous studies reported data from England only or England and Wales; therefore, the results from this study were adjusted for comparison (Appendix A). Since 2000, the total number of licensed DWAA animals in England and Wales has decreased by 68.9%. Farmed species have decreased by 84.9% whereas other species have increased by 59.2%. The number of licences has also fallen from 375 in 2000 to 210 in 2020 (44%). Taxa such as wild cats, crocodilians, and venomous snakes have increased in number since 2000 whilst taxa such as wild boar and ostrich have decreased (Appendix A).

### 3.3. Data Quality

Four local authorities reported current licences for species not on the Schedule, including racoons, crested porcupines, kinkajous, *Boiga dendrophila*, *Ahaetulla nasuta*, *Telescopus semiannulatis,* cotton top tamarins (*Saguinus oedipus*), and squirrel monkeys (*Saimiri sciureus*).

Species-level information was not available for 701 individual animals (4 birds, 530 invertebrates, 141 mammals, 26 reptiles) across 39 licences, while information on genus was not reported for 372 individual animals (314 invertebrates, 49 mammals, 15 reptiles) across 19 licences (excluding hybrids).

Species-level information was lacking for 96.5% of invertebrates.

### 3.4. Licence Costs

Licence fee information was obtained for 359 local licensing authorities (97%). Of these 359 local licensing authorities, 15 were omitted as the vet fee was included in their stated cost, but they did not provide the cost of the vet fee. One licensing authority had no licence fee structure and a further five were omitted as they stated that there was no set fee. This resulted in a sample size of 338, of which 287 (84%) displayed the fee on their website and the remainder required a potential applicant to contact the local licensing authority to obtain the information.

Licence application fees, excluding veterinary fees, ranged from GBP 50.00 to GBP 2102.00. The mean cost of applying for a DWAA licence was GBP 318.49 (*n* = 338). The mean cost of applying for a DWAA licence in a local licensing authority which issued a DWAA licence in 2020 was GBP 285.21 (*n* = 122). The mean cost of applying for a DWAA licence in a local licensing authority which did not issue any DWAA licences in 2020 was GBP 341.08 (*n* = 209). Seven local licensing authorities were omitted from the comparative analysis as it was unknown whether they had issued a licence in 2020.

Data on the cost of licence fees were not normally distributed, and a Wilcoxon–Mann–Whitney test showed that there was a significant association between the cost of licences and issuing licences: local authorities with current DWAA licences (*Mdn* = 250.00), compared to those without current DWAA licences (*Mdn* = 301.00), charged significantly less for a DWAA licence application, W = 15,035, z = −2.72, *p* < 0.01.

Seven local licensing authorities definitively charged an extra fee if additional animals were added to the licence and two of these local licensing authorities also charged extra for additional species. A further two authorities charged extra for additional species but not additional animals.

Three local licensing authorities in the same geographic region reduced the licence application fee by approximately 24% if the DWAA was either a “dwarf caiman” or “serval” (no taxonomic name stated).

Few local licensing authorities could provide a price for the mandatory pre-licence veterinary inspection. Responses from local authorities suggest these vary depending on the amount of time the inspection takes, and the type and number of species visited. Based on the provided fixed veterinary fees, an applicant may expect to pay an average of GBP 299.20 (*n* = 9) for the inspection. Incorporating the estimates and ranges provided, the average veterinary inspection fee was GBP 311.67 (*n* = 15). At least two local licensing authorities expected the applicant to independently seek a veterinary report prior to their application for a DWAA licence. As a result, the average cost of obtaining a DWAA licence in Great Britain in 2021 was estimated to be GBP 630.16.

When adjusted for inflation, the average licensing application costs reported by Greenwood et al. in 2001 were equivalent to GBP 469.24 in 2021. As a result, the average application fee a local authority charges for a Dangerous Wild Animals Act licence are now less (GBP 318.49) than the cost of an equivalent licence twenty years ago.

## 4. Discussion

The results of this study indicate that over one-third of British local licensing authorities had issued a DWAA licence for almost 4000 animals to be kept under 210 licences in 2020. The decreasing number in licences and total animals kept appears to reflect a continued reduction in the keeping of farmed species (bison, boar, and ostrich). In contrast, the keeping of other species has increased to a point where over half of all dangerous wild animals kept under licence could be considered to be non-farm species. With the majority of licensees only keeping one taxonomic class of dangerous wild animal, the results suggest the Act is utilised by a range of hobbyist keepers and potentially some businesses or other establishments too. While these figures are not at a level equivalent to established industries such as licensed pet shops or zoos, the continually changing landscape of dangerous wild animal ownership still makes this area worthy of review [25,26]. The Act’s effectiveness in ensuring human safety and animal welfare, the completeness of the Schedule, and the accuracy of the figures reported all merit consideration in the context of an ever-changing landscape of private animal ownership.

Unfortunately, there is no single publicly available database which records these figures to cross-reference with or aid analysis and review of the Act. Therefore, it is assumed that the information provided by keepers to local licensing authorities and from authorities to the researchers is accurate and up to date. A centralised database would greatly aid any future reviews of the Act.

### 4.1. Changes over Time

The results indicate that the landscape of dangerous wild animal keeping has changed since earlier records, and since the Act came into force. The total number of licensed DWAA animals has fallen substantially since 2000; this is likely to be a result of a decrease in the number of animals kept for farming purposes (bison, boar, and ostrich); plus, other farmed species (guanaco, vicugna (*Vicugna vicugna)*, emus) being removed from the DWAA Schedule in 2007. This trend may have begun earlier: the number of farmed ostriches in the UK has declined from a high of 20,000 individuals in 1995 to 330 individuals in 2020, and wild boar farming grew in Britain from 1981 to the mid-1990s before a subsequent decline [27,28]. Although wild boar comprises the largest population of all taxa in this study, they were kept by just 14 licensees (out of a total of 210). In contrast, the number and proportion of other DWAA animals, including those believed to be kept as exotic pet species, has increased to a point where more are now kept for other purposes than for farming. While farmed species are held in large numbers, but by relatively few individuals, the ownership of other DWAA animals is more widespread.

Licensed keepers generally kept multiple animals of just one taxonomic class of dangerous animal, with mammals as the most kept taxonomic class, both in terms of number of licences and the total number of individuals, even if farmed mammals were removed from the sample.

There have been notable increases in the keeping of certain taxa such as venomous snakes, crocodilians, and wild cats since 2000, indicating that keeping trends may have changed. While it is unclear whether this is a result of increased enforcement of the DWAA or increased popularity in keeping these animals, the rapid development of the internet and online selling platforms since 2000 have almost certainly made the sourcing and acquisition of such animals easier. It is notable that servals and hybrid wild cats represented the most and second-most licensed taxa. The total number of servals and savannah cats licensed now exceed the number of all licensed “small felids” reported by Greenwood et al. [3] and a third of all licensed felids were reported to be hybrids. This is an example of a new animal entering the pet trade, rather than a shift in popularity from one taxon to another, as the savannah was only recognised by The International Cat Association (TICA) in 2001, with its arrival in Britain believed to be around the same time [29]. All of these events took place after Greenwood et al.’s [3] most recent review of the Act. This highlights that trends in popularity of exotic pets are everchanging. Such trends are also now heavily influenced by social media [30,31,32]. This raises questions over the ability of the Act to successfully control surges in demand for fashionable or desirable pets, in particular those which may be missing from the Schedule. The savannah cat’s arrival in Britain and growth in popularity since the last meaningful review of the Act highlights the need for more regular reviews of the Act to keep pace with everchanging pet-keeping trends and ensuring regular and accurate enforcement.

### 4.2. Identification and Taxonomy

The Act relies heavily on accurate and consistent identification of animals to species level. In the Act, S. 7(4) states that the definition “dangerous wild animal” relates to any species stipulated in the first column (scientific name) of the Schedule of the Act. Further, S. 7(5) of the DWAA indicates that common names are for information only, and the classification in the left-hand part of the column should be taken into account in the event of any dispute or legal action [1]. Even during the preliminary debate of the Act, it was noted that, “the difficulty, however, is that common names cannot be relied on alone without introducing serious ambiguities,” and that, “...the only satisfactory way to identify the animals in question is by their scientific classification” [2]. However, binomials and classification may change with taxonomic revision; therefore, an annual review of the Schedule to ensure taxonomy remains up to date and species remain on the Schedule after taxonomic revisions occur is advised.

Disagreements regarding taxonomy can also result in legislative grey areas. The Schedule currently attached to the DWAA highlights that all species of *Casuariidae* require a DWAA licence. Emus are considered within the *Casuariidae* by some sources, notably the International Union for Conservation of Nature, whereas the ITIS classifies emus in the family *Dromaiidae* [33,34]. Emus were specifically removed from the Schedule in 2007; nonetheless, six local licensing authorities stated on their website that a DWAA licence was required for emus in February 2022.

Saki and uacari monkeys are listed under the family *Cebidae* in the Schedule. However, the current accepted taxonomy for this group of monkeys is *Pitheciidae*; a taxonomic family which does not appear within the Schedule of the DWAA [35,36,37]. As a consequence, at least 25 species of saki and uacari monkeys may not be considered dangerous because of taxonomic revision. While saki and uacari monkeys are referenced in the right-hand column of the Schedule, it remains unclear how local authorities would deal with situations involving this family of monkeys as any dispute points towards the left-hand column for resolution. This highlights that the failure to regularly maintain and update the Schedule can result in species being no longer recognised by the Schedule.

Despite the requirement in S. 1(2)(a) of the Act that, “A local authority shall not grant a licence under this Act unless an application for it—(a) specifies the species (whether one or more) of animal, and the number of animals of each species, proposed to be kept under the authority of the licence”, one-fifth of all licences issued in Great Britain contained at least one instance of an animal that did not include the species name. For example, in the case of one local authority, the response was “10 venomous snakes”. When details of species were requested, no additional information could be provided by the licensing authority. In another case, 14 buthid scorpions were licensed, and the local authority was unable to provide details of species. This also raises questions regarding licensee compliance with other related national and international legislation such as CITES and LAIA (where applicable) and how well the Act effectively integrates with the implementation of such legislation. It is unclear whether missing taxonomic information was because the keeper did not know the exact species they were keeping, the keeper had not provided the full details to the local licensing authority, or if the local licensing authority did not provide the correct information to the researchers.

This lack of species information potentially poses a significant risk to public safety, the keeper, and any other individuals within the residence, especially in the case of unidentified venomous animals. While public safety is at the forefront of the Act, the safety of DWAA keepers and others inhabiting the same household are not afforded the same attention. Debating the Dangerous Wild Animals Bill, Lord Newall stated, “We do not mind wild animals tearing their owners to pieces—that is their owners’ affair—but not the general public” [2]. Further, when the Act was modified in 2007, section 7.5 of the explanatory memorandum states, “The focus was on the protection of the general public in the event of escapes, rather than the owner” [8]. Similar species kept under other legislation recommend additional procedures when keeping venomous animals. The Secretary of State’s Standards for Modern Zoo Practice (SSSMZP) recommends a written protocol is in place regarding escapes and bites, medical records of all individuals are available to accompany the person to the hospital, appropriate anti-venoms to be held at the appropriate hospital, and that local medical authorities are made aware of the keeping of venomous species in the area and the protocol for such incidents [4]. The lack of consideration the Act affords to the keeper and others within the keeper’s household appears to be a continued oversight and it is recommended that licence conditions within the Act should consider the health and safety of the household occupants within the setting the animals are kept.

### 4.3. Completeness of the Schedule

There remains no publicly available policy or criteria that define what constitutes a species of wild animal as “dangerous”. When debating the Dangerous Wild Animals Bill, Lord Chelwood stated, “the Bill does not attempt any general definition of a dangerous wild animal, which would indeed be difficult...” [2]. Over twenty years ago, in the last formal examination of the Act, recommendations for an albeit subjective criteria were made and suggested to be based on an animal’s “ferocity x armament” [3]. While the revision to the DWAA in 2007 consulted a “selected group of experts” who particularly looked at the armament and ferocity of species, the harm they could do to a child, the behaviour of the species when cornered or restrained, recorded deaths or serious injury caused by the species, and what legislation already existed to regulate its acquisition or keeping [8]. The continued lack of a definition or clear criteria has resulted in inconsistencies between species included in the Schedule and what is considered to be dangerous in other captive animal legislation or guidance.

Notable exclusions from the current Schedule include venomous varanids, large constrictor snakes, a number of birds of prey species, and several species of deer (*Cervidae*) (Appendix A) [38], especially when compared to Appendix 12—Hazardous animal categorisation within the SSSMZP [4]. Despite their absence from the Schedule, males of certain deer species pose a significant injury risk to humans, especially during the rutting season [39]. Defra [4] also identifies specimens of pythons and boas over 3m as being “greater risk” to the public in licensed zoos. Proposals that the family of snakes known as *Boidae*, which includes such snakes as boa constrictors, pythons, and anacondas, should be added to the Schedule have been put forward in the House of Commons but all previous reviews of the Act have opted against their inclusion in the Schedule [3]. Species are categorised as “greater risk”, “less risk”, and “least risk” in Appendix 12 of the SSSMZP based on an “animal’s likely ferocity and ability to cause harm to people, and the scale of harm if it should do so” [4] where contact between the public and these animals “is likely to cause serious injury or be a serious threat to life, on the basis of hazard and risk of injury, toxin or disease, irrespective of the age and vulnerability of the visitor” [4]. Appendix 12 of the SSSMZP also contains the following note regarding the level of risk of animals, “It should not be interpreted as indicating the level of hazard and risk from animals encountered in any other circumstances. In particular it should not be used to indicate the level of hazard and risk from animals kept in homes, circuses, pet shops and other places not covered by the Act which are subject to the Dangerous Wild Animals Act 1976 for which a separate schedule exists” [4].

Conversely, species removal from the Schedule, such as squirrel monkeys (*Saimiri* spp.), a number of small felids, and the rear-fanged, venomous mangrove snake (*Boiga dendrophila*), seems to have been undertaken on the grounds of a lack of evidence for reported injuries [3,8]. In the case of small primates, comparisons were made between the size of their teeth and those of a domestic cat. A recommendation to include *Boiga irregularis* due to recorded evidence of severe envenomation and human death does not appear to have been implemented [3,40].

It would seem possible that more species pose a greater risk in a domestic setting than within a licensed zoo. Therefore, when considering the Schedule and Appendix 12 of the SSSMZP, it seems unusual that an animal might be considered less dangerous to human safety when kept in a private home or collection than when kept in a licensed zoological garden. The current levels of danger attributed to species under different captive settings appears to lack clear, fundamental rationale in the case of the DWAA. In the short term, the Secretary of State should provide publicly available, clear, and transparent criteria for what determines a species as “dangerous” to aid future reviews of the Act and its Schedule. Meanwhile, it is recommended in the future that the Schedule should be updated to ensure species which pose a threat to the public within other national animal legislation and guidance are added to the Schedule for consistency and ease of enforcement. A contemporary parallel rating system for keeping wild animals in captivity, regardless of the setting, would likely aid understanding, enforcement, and keeper safety.

### 4.4. Animal Welfare

During the Bill’s reading, there were references to animal welfare provision, including, “Before a local authority may grant a licence, it must be satisfied that … the animal’s own welfare will be properly catered for” [2]. However, references to animal welfare within the Act itself are rudimentary and wholly secondary to the provision of public safety. Accommodation must be suitable in regard to environmental, size, and hygiene parameters but also preventing escape, plus food, drink, and bedding must be “adequate and suitable”. There are no requirements relating to behaviour, social groupings, or mental stimulation, nor are there requirements to stipulate the origin of the animal. As previously highlighted, the Scottish Government provides additional guidance on the keeping of dangerous wild animals; however, whether the guidance is suitable can be questioned. For example, the minimum enclosure size for a single lion or tiger is recommended to be 37 m^2^ or less than one-sixth of the size of a tennis court [14,41]. In comparison, a large felid housed in a sanctuary accredited by the Global Federation of Animal Sanctuaries (GFAS) is stated as requiring an outdoor enclosure of at least 111.5m^2^ [42]. The welfare of some DWAA species may also be afforded additional protection under other applicable animal legislation. Previous recommendations from Greenwood et al. [3] stating that public guidance was needed on the minimum standards of care required for commonly kept DWAA scheduled species remain unaddressed. The creation of evidence-based species-specific guidance, species-specific licence conditions for how DWAA species should be housed, and welfare requirements which keepers must meet are recommended.

The results of this study indicate possible welfare issues with primates in particular, in spite of the requirements within the Code of Practice for the Welfare of Privately Kept Non-Human Primates which accompanies the Animal Welfare Act 2006 [15]. The Code of Practice outlines “the steps a keeper of primates must take to meet the needs of an animal as required by Section 9 of the Animal Welfare Act 2006” [15]. Although it is not an offence to breach the requirements, compliance, or lack of, with the Code of Practice can be taken into account by the Court. In spite of the Code of Practice, more than one quarter of licensed primate keepers (12/41) kept single representatives of a primate species. The Code of Practice states, “Primates are long-lived, intelligent, socially complex animals. Being social is a striking feature of primates, and perhaps the most important in terms of meeting their needs. With few exceptions, they live in complex societies that can comprise tens of individual animals” [15]. Guidance by the Scottish Government further outlines the need for all primates to be kept in appropriate social groups [14]. It could be argued that such keepers are in breach of Section 9 of the Animal Welfare Act 2006 [10]. Local Authorities were not challenged on these singularly housed individuals; however, it further highlights the need for mandatory species-specific guidance or enforceable species-specific conditions to be attached to DWAA licences.

### 4.5. Cost

On average, a DWAA licence in 2020 cost 32% less than an equivalent licence in 2000. The cause of this reduction in price is beyond the remit of this research; however, it is possible that technological advances such as online applications that reduce the administrative burden may have contributed.

It appears that the vast majority of local authorities charge a flat fee no matter the number of animals or species on the licence.

Our results indicate a negative association between application cost and presence of licences within local authorities. While local authorities may not impose disproportionate costs to prevent individuals from keeping dangerous wild animals, this could indicate that cost itself and/or associated administration/paperwork may be a factor in preventing potential individuals from keeping or licensing dangerous wild animals [43,44].

Greenwood et al. [3] flagged up the likelihood of non-compliance with the Act. We have no evidence to say whether this has improved. Online sales will have increased since the Act’s last review and certain taxa lend themselves to being kept without licence more than others, particularly small-bodied animals such as reptiles and invertebrates. These animals can be purchased online, shipped or delivered covertly, and kept out of sight of everyone but the household’s occupants [45]. As such, the ease of non-compliance has only increased. This is an area which may require consideration for further research, especially in light of the varying costs of a licence across Britain.

## 5. Conclusions

The Act is predicated upon the assumption that it is possible to keep dangerous wild animals in a way that minimises or eliminates risk to the public and in a manner that does not inherently challenge the welfare of the animals. There is evidence that this assumption may be open to challenge, and as a consequence, the keeping of dangerous animals under licence, at least as currently permitted, may need to be substantially reviewed, restricted, or even ended. The recommendations found throughout are based on areas identified within this study, but also those, where relevant, which were identified by Greenwood et al. [3] over twenty years prior and still remain to be implemented. These aim to resolve problems identified with the Act and its Schedule, plus encourage the creation of improved guidance. If implemented, the recommendations are intended to bring the Act up to date, in line with other national legislation and guidance, and close existing gaps in the legislation which relate to animal welfare and keeper safety.

## Figures and Tables

**Table 1 animals-14-01393-t001:** Total number of local authorities with Dangerous Wild Animals Act (DWAA) licences held within their area, total number of licensees, and total number of individual animals on the DWAA Schedule held within each country of Great Britain.

Country	Local Authorities with Licences	Number of Licensed Keepers	Total Number of Individual Animals
England	114	187	3233
Wales	6	9	462
Scotland	9	14	255
TOTAL	129	210	3950

**Table 2 animals-14-01393-t002:** Minimum, maximum, mean, and median of each DWAA animal taxa kept by licensees in Great Britain. N.B. the total count of licences differs to that in Table 1 due to some DWAA licence holders keeping different taxa within a single licence, e.g., a reptile and a mammal.

	Count of Licences	Min	Max	Mean (µ)	SD	Median	Total Licensed Population
Birds	19	1	189	17.6	42.1	6	334
Invertebrates	8	2	410	68.6	141.8	7	549
Mammals	138	1	400	16.6	44.0	3	2295
Reptiles	61	1	89	12.7	15.9	6	772

## Data Availability

The data that support the findings of this study are available from the corresponding author upon reasonable request.

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
