# Peer review of "Private Keeping of Dangerous Wild Animals in Great Britain"

_animals, 2024, doi:10.3390/ani14101393_

Round 1
Reviewer 1 Report
Comments and Suggestions for Authors
This manuscript provides an updated status of the licensing for dangerous wild animals in the UK, and highlights both changes in trends and opportunities to further clarify or review the Act.
The manuscript is well written, and authors do a good job using data and outcomes to support areas of confusion, lack of consistency, and other areas where the Act can be improved. I believe the information overall is important, particularly with the increasing trend in private ownership of exotic and wild animals.
This survey of licenses and resulting status update appears to be necessary given the prescribed changes in requirements, at both the governmental and local levels. Authors do a good job presenting the effects of these changes, and resultant effects which lead to inconsistencies in process and licensing, confusion, and increased risk to the public. Likewise, Authors note on the animal welfare implications of decreased regulation and inconsistencies in species covered by the Act is supported and appreciated.
I have only one edit: Ln 270. Missing a value for z-score.
The manuscript is well written, cited, and Authors do a good job using data and outcomes to justify their interpretations and recommendations for additional legislative actions to improve the Act and enhance both public and animal safety. I believe the information overall is important, both within the UK and internationally, and particularly relevant given the increasing trend in private ownership of exotic and wild animals.
Author Response
Dear Reviewer 1,
Many thanks for your constructive comments.
The missing statistic has been added to Ln 283. The statistic was re-run in a more up-to-date statistical package (R) and the methodology has been updated accordingly (Ln 180 - 181).
Best wishes
Chris
Reviewer 2 Report
Comments and Suggestions for Authors
This article investigates the scale of dangerous wild animal keeping under licences as regulated by the Dangerous Wild Animals Act 1976, in Great Britain in 2020. It is a review of changes since the initial issuance, limitations of the legislation, discusses issues with species identification, and effectiveness of current legislation in protecting animal welfare and public safety. It is well written and brings to light the extent of the keeping of dangerous wild animals by holders other than certified zoological institutions, circuses, laboratories, or licensed pet sellers.
The manuscript would be enhanced with the addition of a few major topics:
Lines 63-66 lists the requirements for application for a licence. It only specifies information on species of animal and the number of each species to be kept under the licence. What are the criteria for obtaining the licence besides species and numbers? This information should go in the Introduction rather than later in the paper (the Discussion includes some of this information).
For example, in Southwark, UK, their Council website lists the requirements:
https://www.southwark.gov.uk/business/licences/business-premises-licensing/animal-licensing/dangerous-wild-animals#:~:text=You%20must%20have%20a%20license,(PDF%2C%2049kb)%20here.
We'll only grant a licence if:
- the applicant for the licence is over 18 and is a suitable person to hold the licence
- the animal will be kept in secure accommodation so it can't escape
- the accommodation is clean and provides the animal/s with sufficient space for exercise and suitable ventilation, temperature, lighting, drainage
- there are satisfactory precautions in case of a fire (we can ask a local fire officer to inspect the premises)
- adequate precautions are taken to prevent and control infectious diseases
- the applicant is the legal owner of the animal
- the premises have been inspected and approval by a veterinary surgeon has been provided in writing to the local authority
- the licence holder has taken out adequate insurance against liability for any damage caused by the animal
Before granting a licence, we'll require an inspection of the premises by a veterinary surgeon or authorised practitioner.
Is it a requirement to submit the origin of the animals? Are they wild caught, captive born, etc.? Looking at the list of species (Table S1 in supplementary materials), there are many species that are listed as endangered or threatened on the IUCN Red List (CR, EN, VU) which points to the question of how does the DWAA work together with other international or national legislation such as CITES or the Convention on the Conservation of European Wildlife and Natural Habitats (Berne Convention 1982)? The DWAA “recognizes the importance of international efforts to protect endangered species and works in alignment with CITES, including provisions for executing CITES policies”. There should be mention of other legislation that works in conjunction with the DWAA to regulate keeping of these species.
The conclusions note that further review is needed to identify issues and implement resolutions to bring the Act up to date and close existing gaps in the legislation relating to animal welfare and keeper and public safety.
Please access the
comments on the manuscript for specific edits and additional information.

Round 2
Reviewer 2 Report
Comments and Suggestions for Authors
This article investigates the scale of dangerous wild animal keeping under licences as regulated by the Dangerous Wild Animals Act 1976, in Great Britain in 2020. It is a review of changes since the initial issuance, limitations of the legislation, discusses issues with species identification, and effectiveness of current legislation in protecting animal welfare and public safety. It is well written and brings to light the extent of the keeping of dangerous wild animals by holders other than certified zoological institutions, circuses, laboratories, or licensed pet sellers. The authors addressed all the comments on Version 1 and appropriately added the enhancements recommended. These included addition of the criteria needed to apply for a licence to the Introduction rather than in the Discussion and reference to the regulation of endangered or threatened species on the IUCN Red List (CR, EN, VU) and how the DWAA works in conjunction with other international or national legislation such as CITES or the Convention on the Conservation of European Wildlife and Natural Habitats (Berne Convention 1982).
There are a few minor edits:
Line 65 - delete "for a licence"
Line 79 - should be Convention on International Trade (the 'r' is missing in Trade)
Line 81 - CITES 'licences' should be CITES permits
The manuscript was well done and is ready for publication.
